# MAVEN: Multi-Agent Variational Exploration

**Anuj Mahajan**[*†]     **Tabish Rashid**[†]     **Mikayel Samvelyan**[‡]     **Shimon Whiteson**[†]

## Abstract

Centralised training with decentralised execution is an important setting for cooperative deep multi-agent reinforcement learning due to communication constraints during execution and computational tractability in training. In this paper, we analyse value-based methods that are known to have superior performance in complex environments [43]. We specifically focus on QMIX [40], the current state-of-the-art in this domain. We show that the representational constraints on the joint action-values introduced by QMIX and similar methods lead to provably poor exploration and suboptimality. Furthermore, we propose a novel approach called MAVEN that hybridises value and policy-based methods by introducing a latent space for hierarchical control. The value-based agents condition their behaviour on the shared latent variable controlled by a hierarchical policy. This allows MAVEN to achieve committed, temporally extended exploration, which is key to solving complex multi-agent tasks. Our experimental results show that MAVEN achieves significant performance improvements on the challenging SMAC domain [43].

## 1 Introduction

Cooperative *multi-agent reinforcement learning* (MARL) is a key tool for addressing many real-world problems such as coordination of robot swarms [22] and autonomous cars [6]. However, two key challenges stand between cooperative MARL and such real-world applications. First, scalability is limited by the fact that the size of the joint action space grows exponentially in the number of agents. Second, while the training process can typically be centralised, partial observability and communication constraints often mean that execution must be decentralised, i.e., each agent can condition its actions only on its local action-observation history, a setting known as *centralised training with decentralised execution* (CTDE).

While both policy-based [13] and value-based [40, 48, 46] methods have been developed for CTDE, the current state of the art, as measured on SMAC, a suite of StarCraft II micromanagement benchmark tasks [43], is a value-based method called QMIX [40]. QMIX tries to address the challenges mentioned above by learning *factored* value functions. By decomposing the joint value function into factors that depend only on individual agents, QMIX can cope with large joint action spaces. Furthermore, because such factors are combined in a way that respects a monotonicity constraint, each agent can select its action based only on its own factor, enabling decentralised execution. However, this decentralisation comes with a price, as the monotonicity constraint restricts QMIX to suboptimal value approximations.

QTRAN[44], another recent method, performs this trade-off differently by formulating multi-agent learning as an optimisation problem with linear constraints and relaxing it with $L2$ penalties for tractability.

In this paper, we shed light on a problem unique to decentralised MARL that arises due to inefficient exploration. Inefficient exploration hurts decentralised MARL, not only in the way it hurts single agent

---

[*]Correspondence to: `anuj.mahajan@cs.ox.ac.uk`

[†]Dept. of Computer Science, University of Oxford

[‡]Russian-Armenian University

RL[33] (by increasing sample inefficiency[31, 30]), but also by interacting with the representational constraints necessary for decentralisation to push the algorithm towards suboptimal policies. Single agent RL can avoid convergence to suboptimal policies using various strategies like increasing the exploration rate ($\epsilon$) or policy variance, ensuring optimality in the limit. However, we show, both theoretically and empirically, that the same is not possible in decentralised MARL.

Furthermore, we show that *committed* exploration can be used to solve the above problem. In committed exploration [36], exploratory actions are performed over extended time steps in a coordinated manner. Committed exploration is key even in single-agent exploration but is especially important in MARL, as many problems involve long-term coordination, requiring exploration to discover temporally extended joint strategies for maximising reward. Unfortunately, none of the existing methods for CTDE are equipped with *committed* exploration.

To address these limitations, we propose a novel approach called *multi-agent variational exploration* (MAVEN) that hybridises value and policy-based methods by introducing a latent space for hierarchical control. MAVEN's value-based agents condition their behaviour on the shared latent variable controlled by a hierarchical policy. Thus, fixing the latent variable, each joint action-value function can be thought of as a mode of joint exploratory behaviour that persists over an entire episode. Furthermore, MAVEN uses mutual information maximisation between the trajectories and latent variables to learn a diverse set of such behaviours. This allows MAVEN to achieve committed exploration while respecting the representational constraints. We demonstrate the efficacy of our approach by showing significant performance improvements on the challenging SMAC domain.

## 2 Background

We model the fully cooperative multi-agent task as a Dec-POMDP [34], which is formally defined as a tuple $G = \langle S, U, P, r, Z, O, n, \gamma \rangle$. $S$ is the state space of the environment. At each time step $t$, every agent $i \in \mathcal{A} \equiv \{1, ..., n\}$ chooses an action $u^i \in U$ which forms the joint action $\mathbf{u} \in \mathbf{U} \equiv U^n$. $P(s'|s, \mathbf{u}) : S \times \mathbf{U} \times S \to [0, 1]$ is the state transition function. $r(s, \mathbf{u}) : S \times \mathbf{U} \to \mathbb{R}$ is the reward function shared by all agents and $\gamma \in [0, 1)$ is the discount factor. We consider *partially observable* settings, where each agent does not have access to the full state and instead samples observations $z \in Z$ according to observation function

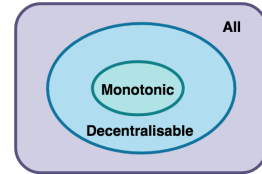

Figure 1: Classification of MARL problems.

$O(s, i) : S \times \mathcal{A} \to Z$. The action-observation history for an agent $i$ is $\tau^i \in T \equiv (Z \times U)^*$ on which it can condition its policy $\pi^i(u^i|\tau^i) : T \times U \to [0, 1]$. We use $u^{-i}$ to denote the action of all the agents other than $i$ and follow a similar convention for the policies $\pi^{-i}$. The joint policy $\pi$ is based on *action-value function*: $Q^\pi(s_t, \mathbf{u}_t) = \mathbb{E}_{s_{t+1:\infty}, \mathbf{u}_{t+1:\infty}} \left[ \sum_{k=0}^{\infty} \gamma^k r_{t+k} | s_t, \mathbf{u}_t \right]$. The goal of the problem is to find the optimal action value function $Q^*$. During centralised training, the learning algorithm has access to the action-observation histories of all agents and the full state. However, each agent can only condition on its own local action-observation history $\tau^i$ during decentralised execution (hence the name CTDE). For CTDE methods factoring action values over agents, we represent the individual agent utilities by $q_i, i \in \mathcal{A}$. An important concept for such methods is *decentralisability* (see IGM in [44]) which asserts that $\exists q_i$, such that $\forall s, \mathbf{u}$:

$$\arg\max_{\mathbf{u}} Q^*(s, \mathbf{u}) = \left( \arg\max_{u^1} q_1(\tau^1, u^1) \ldots \arg\max_{u^n} q_n(\tau^n, u^n) \right)', \tag{1}$$

Fig. 1 gives the classification for MARL problems. While the containment is strict for partially observable setting, it can be shown that all tasks are decentralisable given full observability and sufficient representational capacity.

**QMIX [40]** is a value-based method that learns a monotonic approximation $Q_{qmix}$ for the joint action-value function. Figure 8 in Appendix A illustrates its overall setup, reproduced for convenience. QMIX factors the joint-action $Q_{qmix}$ into a monotonic nonlinear combination of individual utilities $q_i$ of each agent which are learnt via a *utility network*. A *mixer network* with nonnegative weights is responsible for combining the agent's utilities for their chosen actions $u^i$ into $Q_{qmix}(s, \mathbf{u})$. This nonnegativity ensures that $\frac{\partial Q_{qmix}(s, \mathbf{u})}{\partial q_i(s, u^i)} \geq 0$, which in turn guarantees Eq. (1). This decomposition allows for an efficient, tractable maximisation as it can be performed in $\mathcal{O}(n|U|)$ time as opposed to $\mathcal{O}(|U|^n)$. Additionally, it allows for easy decentralisation as each agent can independently perform

an argmax. During learning, the QMIX agents use $\epsilon$-greedy exploration over their individual utilities to ensure sufficient exploration. For VDN [46] the factorization is further restrained to be just the sum of utilities: $Q_{vdn}(s, \mathbf{u}) = \sum_i q_i(s, u^i)$.

**QTRAN [44]** is another value-based method. Theorem 1 in the QTRAN paper guarantees optimal decentralisation by using linear constraints between agent utilities and joint action values, but it imposes $\mathcal{O}(|S||U|^n)$ constraints on the optimisation problem involved, where $|\cdot|$ gives set size. This is computationally intractable to solve in discrete state-action spaces and is impossible given continuous state-action spaces. The authors propose two algorithms (QTRAN-base and QTRAN-alt) which relax these constraints using two L2 penalties. While QTRAN tries avoid QMIX's limitations, we found that it performs poorly in practice on complex MARL domains (see Section 5) as it deviates from the exact solution due to these relaxations.

## 3  Analysis

In this section, we analyse the policy learnt by QMIX in the case where it cannot represent the true optimal action-value function. Our analysis is not restricted to QMIX and can easily be extended to similar algorithms like VDN [46] whose representation class is a subset of QMIX. Intuitively, monotonicity implies that the optimal action of agent $i$ does not depend on the actions of the other agents. This motivates us to characterise the class of $Q$-functions that cannot be represented by QMIX, which we call *nonmonotonic Q* functions.

**Definition 1** (Nonmonotonicity). *For any state $s \in S$ and agent $i \in \mathcal{A}$ given the actions of the other agents $u^{-i} \in U^{n-1}$, the $Q$-values $Q(s, (u^i, u^{-i}))$ form an ordering over the action space of agent $i$. Define $C(i, u^{-i}) := \{(u_1^i, ..., u_{|U|}^i) | Q(s, (u_j^i, u^{-i})) \geq Q(s, (u_{j+1}^i, u^{-i})), j \in \{1, \ldots, |U|\}, u_j^i \in U, j \neq j' \implies u_j^i \neq u_{j'}^i\}$, as the set of all possible such orderings over the action-values. The joint-action value function is **nonmonotonic** if $\exists i \in \mathcal{A}, u_1^{-i} \neq u_2^{-i}$ s.t. $C(i, u_1^{-i}) \cap C(i, u_2^{-i}) = \varnothing$.*

A simple example of a nonmonotonic $Q$-function is given by the payoff matrix of the two-player three-action matrix game shown on Table 1(a). Table 1(b) shows the values learned by QMIX under *uniform visitation*, i.e., when all state-action pairs are explored equally.

|   | A | B | C |   |   | A | B | C |
|---|---|---|---|---|---|---|---|---|
| A | 10.4 | 0 | 10 |   | A | 6.08 | 6.08 | 8.95 |
| B | 0 | 10 | 10 |   | B | 6.00 | 5.99 | 8.87 |
| C | 10 | 10 | 10 |   | C | 8.99 | 8.99 | 11.87 |
|   | (a) |   |   |   |   | (b) |   |   |

Table 1: (a) An example of a nonmonotonic payoff matrix, (b) QMIX values under uniform visitation.

Of course, the fact that QMIX cannot represent the optimal value function does not imply that the policy it learns must be suboptimal. However, the following analysis establishes the suboptimality of such policies.

**Theorem 1** (Uniform visitation). *For $n$-player, $k \geq 3$-action matrix games ($|\mathcal{A}| = n, |U| = k$), under uniform visitation, $Q_{qmix}$ learns a $\delta$-suboptimal policy for any time horizon $T$, for any $0 < \delta \leq R\left[\sqrt{\frac{a(b+1)}{a+b}} - 1\right]$ for the payoff matrix ($n$-dimensional) given by the template below, where $b = \sum_{s=1}^{k-2}\binom{n+s-1}{s}$, $a = k^n - (b+1)$, $R > 0$:*

$$
\begin{bmatrix}
R + \delta & 0 & \ldots & R \\
0 & & & \iddots \\
\vdots & \iddots & & \vdots \\
R & \ldots & & R
\end{bmatrix}
$$

*Proof.* see Appendix B.1 □

We next consider $\epsilon$-greedy visitation, in which each agent uses an $\epsilon$-greedy policy and $\epsilon$ decreases over time. Below we provide a probabilistic bound on the maximum possible value of $\delta$ for QMIX to learn a suboptimal policy for any time horizon $T$.

**Theorem 2** ($\epsilon$-greedy visitation). *For n-player, $k \geq 3$-action matrix games, under $\epsilon$-greedy visitation $\epsilon(t)$, $Q_{qmix}$ learns a $\delta$-suboptimal policy for any time horizon $T$ with probability*

$$\geq 1 - \left( \exp(-\tfrac{Tv^2}{2}) + (k^n - 1) \exp(-\tfrac{Tv^2}{2(k^n-1)^2}) \right), \text{ for any } 0 < \delta \leq R\left[ \sqrt{a\left( \tfrac{vb}{2(1-v/2)(a+b)} + 1 \right)} - 1 \right]$$

*for the payoff matrix given by the template above, where $b = \sum_{s=1}^{k-2} \binom{n+s-1}{s}$, $a = k^n - (b+1)$, $R > 0$ and $v = \epsilon(T)$.*

*Proof.* see Appendix B.2 □

The reliance of QMIX on $\epsilon$-greedy action selection prevents it from engaging in committed exploration [36], in which a precise sequence of actions must be chosen in order to reach novel, interesting parts of the state space. Moreover, Theorems 1 and 2 imply that the agents can latch onto suboptimal behaviour early on, due to the monotonicity constraint. Theorem 2 in particular provides a surprising result: For a fixed time budget $T$, increasing QMIX's exploration rate lowers its probability of learning the optimal action due to its representational limitations. Intuitively this is because the monotonicity constraint can prevent the $Q$-network from correctly remembering the true value of the optimal action (currently perceived as suboptimal). We hypothesise that the lack of a principled exploration strategy coupled with these representational limitations can often lead to catastrophically poor exploration, which we confirm empirically.

## 4 Methodology

In this section, we propose *multi-agent variational exploration* (MAVEN), a new method that overcomes the detrimental effects of QMIX's monotonicity constraint on exploration. MAVEN does so by learning a diverse ensemble of monotonic approximations with the help of a latent space. Its architecture consists of value-based agents that condition their behaviour on the shared latent variable $z$ controlled by a hierarchical policy that offloads $\epsilon$-greedy with committed exploration. Thus, fixing $z$, each joint action-value function is a monotonic approximation to the optimal action-value function that

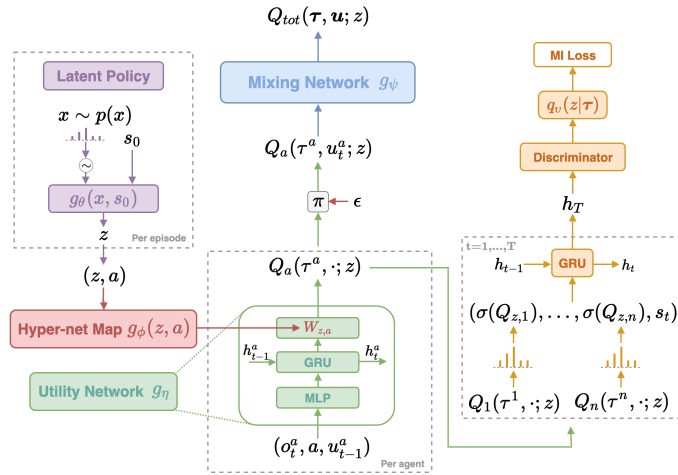

Figure 2: Architecture for MAVEN.

is learnt with $Q$-learning. Furthermore, each such approximation can be seen as a mode of committed joint exploratory behaviour. The latent policy over $z$ can then be seen as exploring the space of *joint behaviours* and can be trained using any policy learning method. Intuitively, the $z$ space should map to diverse modes of behaviour. Fig. 2 illustrates the complete setup for MAVEN. We first focus on the lefthand side of the diagram, which describes the learning framework for the latent space policy and the joint action values. We parametrise the hierarchical policy by $\theta$, the agent utility network with $\eta$, the hypernet map from latent variable $z$ used to condition utilities by $\phi$, and the mixer net with $\psi$. $\eta$ can be associated with a feature extraction module per agent and $\phi$ can be associated with the task of modifying the utilities for a particular mode of exploration. We model the hierarchical policy $\pi_z(\cdot|s_0; \theta)$ as a transformation of a simple random variable $x \sim p(x)$ through a neural network parameterised by $\theta$; thus $z \sim g_\theta(x, s_0)$, where $s_0$ is initial state. Natural choices for $p(x)$ are uniform for discrete $z$ and uniform or normal for continuous $z$.

We next provide a coordinate ascent scheme for optimising the parameters. Fixing $z$ gives a joint action-value function $Q(\mathbf{u}, s; z, \phi, \eta, \psi)$ which implicitly defines a greedy deterministic policy $\pi_{\mathcal{A}}(\mathbf{u}|s; z, \phi, \eta, \psi)$ (we drop the parameter dependence wherever its inferable for clarity of presentation). This gives the corresponding $Q$-learning loss:

$$\mathcal{L}_{QL}(\phi, \eta, \psi) = \mathbb{E}_{\pi_{\mathcal{A}}}[(Q(\mathbf{u}_t, s_t; z) - [r(\mathbf{u}_t, s_t) + \gamma \max_{\mathbf{u}_{t+1}} Q(\mathbf{u}_{t+1}, s_{t+1}; z)])^2],$$

where $t$ is the time step. Next, fixing $\phi, \eta, \psi$, the hierarchical policy over $\pi_z(\cdot|s_0; \theta)$ is trained on the cumulative trajectory reward $\mathcal{R}(\boldsymbol{\tau}, z|\phi, \eta, \psi) = \sum_t r_t$ where $\boldsymbol{\tau}$ is the joint trajectory.

---

**Algorithm 1** MAVEN

Initialize parameter vectors $\upsilon, \phi, \eta, \psi, \theta$
Learning rate $\leftarrow \alpha, \mathcal{D} \leftarrow \{\}$
**for** each episodic iteration i **do**
  $s_0 \sim \rho(s_0), x \sim p(x), z \sim g_\theta(x; s_0)$
  **for** each environment step t **do**
    $\mathbf{u}_t \sim \pi_{\mathcal{A}}(u|s_t; z, \phi, \eta, \psi)$
    $s_{t+1} \sim p(s_{t+1}|s_t, \mathbf{u}_t)$
    $\mathcal{D} \leftarrow \mathcal{D} \cup \{(s_t, \mathbf{u}_t, r(s_t, \mathbf{u}_t), r^z_{aux}(\boldsymbol{\tau}_i), s_{t+1})\}$
  **end for**
  **for** each gradient step **do**
    $\phi \leftarrow \phi + \alpha \hat{\nabla}_\phi(\lambda_{MI}\mathcal{J}_V - \lambda_{QL}\mathcal{L}_{QL})$ (Hypernet update)
    $\eta \leftarrow \eta + \alpha \hat{\nabla}_\eta(\lambda_{MI}\mathcal{J}_V - \lambda_{QL}\mathcal{L}_{QL})$ (Feature update)
    $\psi \leftarrow \psi + \alpha \hat{\nabla}_\psi(\lambda_{MI}\mathcal{J}_V - \lambda_{QL}\mathcal{L}_{QL})$ (Mixer update)
    $\upsilon \leftarrow \upsilon + \alpha \hat{\nabla}_\upsilon \lambda_{MI}\mathcal{J}_V$ (Variational update)
    $\theta \leftarrow \theta + \alpha \hat{\nabla}_\theta \mathcal{J}_{RL}$ (Latent space update)
  **end for**
**end for**

---

Thus, the hierarchical policy objective for $z$, freezing the parameters $\psi, \eta, \phi$ is given by:

$$\mathcal{J}_{RL}(\theta) = \int \mathcal{R}(\tau_{\mathcal{A}}|z)p_\theta(z|s_0)\rho(s_0)dzds_0.$$

However, the formulation so far does not encourage diverse behaviour corresponding to different values of $z$ and all the values of $z$ could collapse to the same joint behaviour. To prevent this, we introduce a *mutual information* (MI) objective between the observed trajectories $\boldsymbol{\tau} \triangleq \{(\mathbf{u}_t, s_t)\}$, which are representative of the joint behaviour and the latent variable $z$. The actions $\mathbf{u}_t$ in the trajectory are represented as a stack of agent utilities and $\sigma$ is an operator that returns a per-agent Boltzmann policy w.r.t. the utilities at each time step $t$, ensuring the MI objective is differentiable and helping train the network parameters $(\psi, \eta, \phi)$. We use an RNN [20] to encode the entire trajectory and then maximise $MI(\sigma(\boldsymbol{\tau}), z)$. Intuitively, the MI objective encourages visitation of diverse trajectories $\boldsymbol{\tau}$ while at the same time making them identifiable given $z$, thus elegantly separating the $z$ space into different exploration modes. The MI objective is:

$$\mathcal{J}_{MI} = \mathcal{H}(\sigma(\boldsymbol{\tau})) - \mathcal{H}(\sigma(\boldsymbol{\tau})|z) = \mathcal{H}(z) - \mathcal{H}(z|\sigma(\boldsymbol{\tau})),$$

where $\mathcal{H}$ is the entropy. However, neither the entropy of $\sigma(\boldsymbol{\tau})$ nor the conditional of $z$ given the former is tractable for nontrivial mappings, which makes directly using MI infeasible. Therefore, we introduce a variational distribution $q_\upsilon(z|\sigma(\boldsymbol{\tau}))$ [50, 3] parameterised by $\upsilon$ as a proxy for the posterior over $z$, which provides a lower bound on $\mathcal{J}_{MI}$ (see Appendix B.3).

$$\mathcal{J}_{MI} \geq \mathcal{H}(z) + \mathbb{E}_{\sigma(\boldsymbol{\tau}), z}[\log(q_\upsilon(z|\sigma(\boldsymbol{\tau})))].$$

We refer to the righthand side of the above inequality as the variational MI objective $\mathcal{J}_V(\upsilon, \phi, \eta, \psi)$. The lower bound matches the exact MI when the variational distribution equals $p(z|\sigma(\boldsymbol{\tau}))$, the true posterior of $z$. The righthand side of Fig. 2 gives the network architectures corresponding to the variational MI loss. Since

$$\mathbb{E}_{\boldsymbol{\tau}, z}[\log(q_\upsilon(z|\sigma(\cdot)))] = \mathbb{E}_{\boldsymbol{\tau}}[-KL(p(z|\sigma(\cdot))||q_\upsilon(z|\sigma(\cdot))] - \mathcal{H}(z|\sigma(\cdot)),$$

where the nonnegativity of the KL divergence on the righthand side implies that a bad variational approximation can hurt performance as it induces a gap between the true objective and the lower bound [32, 2]. This problem is especially important if $z$ is chosen to be continuous as for discrete distributions the posterior can be represented exactly as long as the dimensionality of $\upsilon$ is greater than the number of categories for $z$. The problem can be addressed by various state-of-the-art developments in amortised variational inference [42, 41]. The variational approximation can also be seen as a discriminator/critic that induces an auxiliary reward field $r_{aux}^z(\boldsymbol{\tau}) = \log(q_\upsilon(z|\sigma(\boldsymbol{\tau}))) - \log(p(z))$ on the trajectory space. Thus the overall objective becomes:

$$\max_{\upsilon,\phi,\eta,\psi,\theta} \mathcal{J}_{RL}(\theta) + \lambda_{MI}\mathcal{J}_V(\upsilon,\phi,\eta,\psi) - \lambda_{QL}\mathcal{L}_{QL}(\phi,\eta,\psi),$$

where $\lambda_{MI}, \lambda_{QL}$ are positive multipliers. For training (see Algorithm 1), at the beginning of each episode we sample an $x$ and obtain $z$ and then unroll the policy until termination and train $\psi, \eta, \phi, \upsilon$ on the $Q$-learning loss corresponding to greedy policy for the current exploration mode and the variational MI reward. The hierarchical policy parameters $\theta$ can be trained on the true task return using any policy optimisation algorithm. At test time, we sample $z$ at the start of an episode and then perform a decentralised argmax on the corresponding $Q$-function to select actions. Thus, MAVEN achieves committed exploration while respecting QMIX's representational constraints.

## 5 Experimental Results

We now empirically evaluate MAVEN on various new and existing domains.

### 5.1 m-step matrix games

To test the how nonmonotonicity and exploration interact, we introduce a simple $m$-step matrix game. The initial state is nonmonotonic, zero rewards lead to termination, and the differentiating states are located at the terminal ends; there are $m - 2$ intermediate states. Fig. 3(a) illustrates the $m$-step matrix game for $m = 10$. The optimal policy is to take the top left joint action and finally take the bottom right action, giving an optimal total payoff of $m + 3$. As $m$ increases, it becomes increasingly difficult to discover the optimal policy using $\epsilon$-dithering and a *committed* approach becomes necessary. Additionally, the initial state's nonmonotonicity provides inertia against switching the policy to the other direction. Fig. 3(b) plots median returns for $m = 10$. QMIX gets stuck in a suboptimal policy with payoff 10, while MAVEN successfully learns the true optimal policy with payoff 13. This example shows how representational constraints can hurt performance if they are left unmoderated.

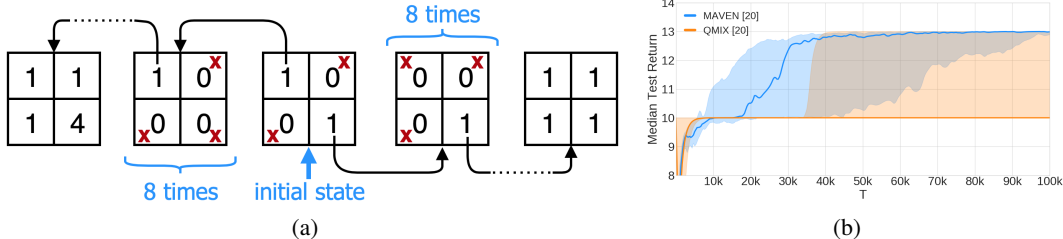

(a)                                        (b)

Figure 3: (a) $m$-step matrix game for $m = 10$ case (b) median return of MAVEN and QMIX method on 10-step matrix game for 100k training steps, averaged over 20 random initializations (2nd and 3rd quartile is shaded).

### 5.2 StarCraft II

**StarCraft Multi-Agent Challenge**   We consider a challenging set of cooperative StarCraft II maps from the SMAC benchmark [43] which Samvelyan et al. have classified as **Easy, Hard** and **Super Hard**. Our evaluation procedure is similar to [40, 43]. We pause training every 100000 time steps and run 32 evaluation episodes with decentralised greedy action selection. After training, we report the median *test win rate* (percentage of episodes won) along with 2nd and 3rd quartiles (shaded in plots). We use grid search to tune hyperparameters. Appendix C.1 contains additional experimental details. We compare MAVEN, QTRAN, QMIX, COMA [13] and IQL [48] on several SMAC maps. Here we present the results for two **Super Hard** maps `corridor` & `6h_vs_8z` and an **Easy** map `2s3z`. The `corridor` map, in which 6 Zealots face 24 enemy Zerglings, requires agents to make effective use of the terrain features and block enemy attacks from different directions. A properly

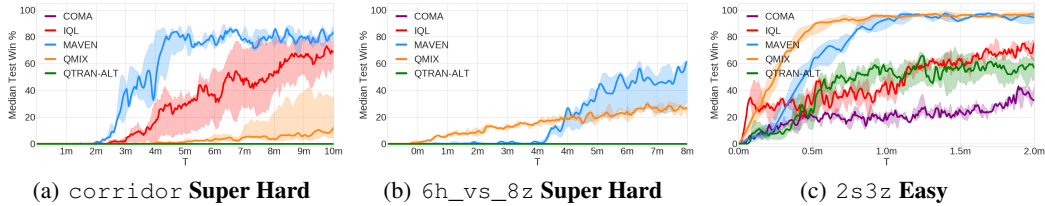

(a) corridor **Super Hard**    (b) 6h_vs_8z **Super Hard**    (c) 2s3z **Easy**

Figure 4: The performance of various algorithms on three SMAC maps.

*coordinated* exploration scheme applied to this map would help the agents discover a suitable unit positioning quickly and improve performance. 6h_vs_8z requires fine grained 'focus fire' by the allied Hydralisks. 2s3z requires agents to learn "focus fire" and interception. Figs. 4(a) to 4(c) show the median win rates for the different algorithms on the maps; additional plots can be found in Appendix C.2. The plots show that MAVEN performs substantially better than all alternate approaches on the **Super Hard** maps with performance similar to QMIX on **Hard** and **Easy** maps. Thus MAVEN performs better as difficulty increases. Furthermore, QTRAN does not yield satisfactory performance on most SMAC maps ($0\%$ win rate). The map on which it performs best is 2s3z (Fig. 4(c)), an **Easy** map, where it is still worse than QMIX and MAVEN. We believe this is because QTRAN enforces decentralisation using only relaxed L2 penalties that are insufficient for challenging domains.

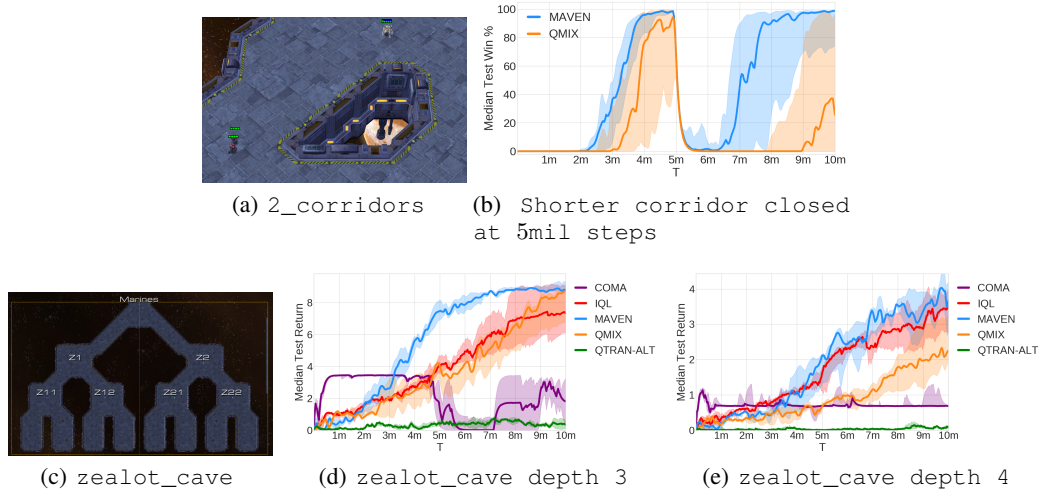

(a) 2_corridors    (b) Shorter corridor closed at 5mil steps

(c) zealot_cave    (d) zealot_cave depth 3    (e) zealot_cave depth 4

Figure 5: State exploration and policy robustness

**Exploration and Robustness**    Although SMAC domains are challenging, they are not specially designed to test state-action space exploration, as the units involved start engaging immediately after spawning. We thus introduce a new SMAC map designed specifically to assess the effectiveness of multi-agent exploration techniques and their ability to adapt to changes in the environment. The 2-corridors map features two Marines facing an enemy Zealot. In the beginning of training, the agents can make use of two corridors to attack the enemy (see Fig. 5(a)). Halfway through training, the short corridor is blocked. This requires the agents to adapt accordingly and use the long corridor in a coordinated way to attack the enemy. Fig. 5(b) presents the win rate for MAVEN and QMIX for 2-corridors when the gate to short corridor is closed after 5 million steps. While QMIX fails to recover after the closure, MAVEN swiftly adapts to the change in the environment and starts using the long corridor. MAVEN's latent space allows it to explore in a *committed* manner and associate use of the long corridor with a value of $z$. Furthermore, it facilitates recall of the behaviour once the short corridor becomes unavailable, which QMIX struggles with due to its representational constraints. We also introduce another new map called zealot_cave to test state exploration, featuring a tree-structured cave with a Zealot at all but the leaf nodes (see Fig. 5(c)). The agents consist of 2 marines who need to learn 'kiting' to reach all the way to the leaf nodes and get extra reward only if they always take the right branch except at the final intersection. The depth of the cave offers control over the task difficulty. Figs. 5(d) and 5(e) give the average reward received by the different algorithms for cave depths of 3 and 4. MAVEN outperforms all algorithms compared.

**Representability** The optimal action-value function lies outside of the representation class of the CTDE algorithm used for most interesting problems. One way to tackle this issue is to find local approximations to the optimal value function and choose the best local approximation given the observation. We hypothesise that MAVEN enables application of this principle by mapping the latent space

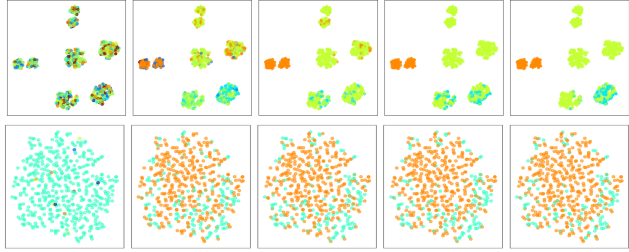

Figure 6: tsne plot for $s_0$ labelled with $z$ (16 categories), initial (left) to final (right), top `3s5z`, bottom `micro_corridor`

$z$ to local approximations and using the hierarchical policy to choose the best such approximation given the initial state $s_0$, thus offering better representational capacity while respecting the constraints requiring decentralization. To demonstrate this, we plot the t-SNE [29] of the initial states and colour them according to the latent variable sampled for it using the hierarchical policy at different time steps during training. The top row of Fig. 6 gives the time evolution of the plots for `3s5z` which shows that MAVEN learns to associate the initial state clusters with the same latent value, thus partitioning the state-action space with distinct *joint behaviours*. Another interesting plot in the bottom row for `micro_corridor` demonstrates how MAVEN's latent space allows transition to more rewarding joint behaviour which existing methods would struggle to accomplish.

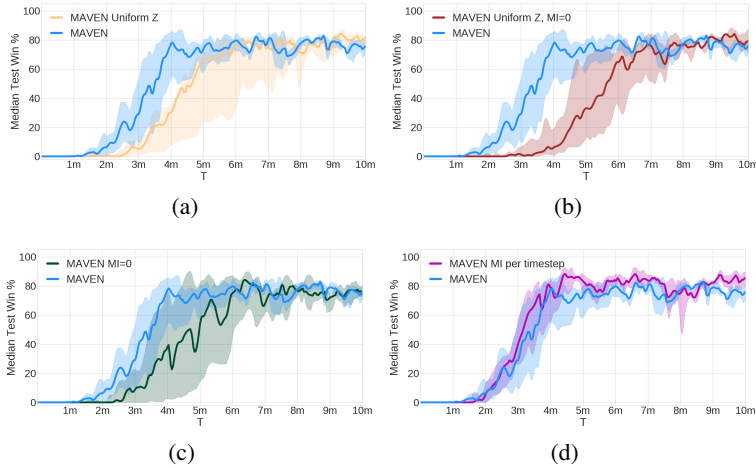

Figure 7: (a) & (b) investigate uniform hierarchical policy. (c) & (d) investigate effects of MI loss.

**Ablations** We perform several ablations on the `micro_corridor` scenario with $Z = 16$ to try and determine the importance of each component of MAVEN. We first consider using a fixed uniform hierarchical policy over $z$. Fig. 7(a) shows that MAVEN with a uniform policy over $z$ performs worse than a learned policy. Interestingly, using a uniform hierarchical policy and no variational MI loss to encourage diversity results in a further drop in performance, as shown in Fig. 7(b). Thus sufficient diversification of the observed trajectories via an explicit agency is important to find good policies ensuring sample efficiency. Fig. 7(b) is similar to Bootstrapped-DQN [36], which has no incentive to produce diverse behaviour other than the differing initialisations depending on $z$. Thus, all the latent variable values can collapse to the same *joint behaviour*. If we are able to learn a hierarchical policy over $z$, we can focus our computation and environmental samples on the more promising variables, which allows for better final performance. Fig. 7(c) shows improved performance relative to Fig. 7(b) providing some evidence for this claim. Next, we consider how the different choices of variational MI loss (*per time step, per trajectory*) affect performance in Fig. 7(d). Intuitively, the per time step loss promotes a more spread out exploration as it forces the discriminator to learn the inverse map to the latent variable at each step. It thus tends to distribute its exploration budget at each step uniformly, whereas the trajectory loss allows the *joint behaviours* to be similar for extended durations and take diversifying actions at only a few time steps in a trajectory, keeping its spread fairly narrow. However, we found that in most scenarios, the two losses perform similarly. See Appendix C.2 for additional plots and ablation results.

# 6 Related Work

In recent years there has been considerable work extending MARL from small discrete state spaces that can be handled by tabular methods [51, 5] to high-dimensional, continuous state spaces that require the use of function approximators [13, 28, 39]. To tackle computational intractability from exponential blow-up of state-action space, Guestrin et al. [16, 17] use coordination graphs to factor large MDPs for multi-agent systems and propose inter-agent communication arising from message passing on the graphs. Similarly [45, 11, 23] model inter-agent communication explicitly. In CTDE [26], [47] extend Independent $Q$-Learning [48] to use DQN to learn $Q$-values for each agent independently. [12, 35] tackle the instability that arises from training the agents independently. Lin et al.[27] first learn a centralised controller to solve the task, and then train the agents to imitate its behaviour. Sunehag et al. [46] propose *Value Decomposition Networks* (VDN), which learn the joint-action $Q$-values by factoring them as the sum of each agent's $Q$-values. QMIX [40] extends VDN to allow the joint action $Q$-values to be a monotonic combination of each agent's $Q$-Values that can vary depending on the state. Section 4 outlines how MAVEN builds upon QMIX. QTRAN [44] approaches the suboptimality vs. decentralisation tradeoff differently by introducing relaxed L2 penalties in the RL objective. [15] maximise the *empowerment* between one agents actions and the others future state in a competitive setting. Zheng et al. [52] allow each agent to condition their policies on a shared continuous latent variable. In contrast to our setting, they consider the fully-observable centralised control setting and do not attempt to enforce diversity across the shared latent variable. Aumann [1] proposes the concept of a *correlated* equilibrium in non-cooperative multi-agent settings in which each agent conditions its policy on some shared variable that is sampled every episode.

In the single agent setting, Osband et al.[36] learn an ensemble of $Q$-value functions (which all share weights except for the final few layers) that are trained on their own sampled trajectories to approximate a posterior over $Q$-values via the statistical bootstrapping method. MAVEN without the MI loss and a uniform policy over $z$ is then equivalent to each agent using a Bootstrapped DQN. [37] extends the Bootstrapped DQN to include a prior. [7] consider the setting of *concurrent* RL in which multiple agents interact with their own environments in parallel. They aim to achieve more efficient exploration of the state-action space by seeding each agent's parametric distributions over MDPs with different seeds, whereas MAVEN aims to achieve this by maximising the mutual information between $z$ and a trajectory.

Yet another direction of related work lies in defining intrinsic rewards for single agent hierarchical RL that enable learning of diverse behaviours for the low level layers of the hierarchical policy. Florensa et al. [10] use hand designed state features and train the lower layers of the policy by maximising MI, and then tune the policy network's upper layers for specific tasks. Similarly [14, 8] learn a mixture of diverse behaviours using deep neural networks to extract state features and use MI maximisation between them and the behaviours to learn useful skills without a reward function. MAVEN differs from DIAYN [8] in the use case, and also enforces *action diversification* due to MI being maximised jointly with states and actions in a trajectory. Hence, agents jointly learn to solve the task is many different ways; this is how MAVEN prevents suboptimality from representational constraints, whereas DIAYN is concerned only with discovering new states. Furthermore, DIAYN trains on diversity rewards using RL whereas we train on them via gradient ascent. Haarnoja et al. [18] use normalising flows [41] to learn hierarchical latent space policies using max entropy RL [49, 53, 9], which is related to MI maximisation but ignores the variational posterior over latent space behaviours. In a similar vein [21, 38] use auxiliary rewards to modify the RL objective towards a better tradeoff between exploration and exploitation.

# 7 Conclusion and Future work

In this paper, we analysed the effects of representational constraints on exploration under CTDE. We also introduced MAVEN, an algorithm that enables committed exploration while obeying such constraints. As immediate future work, we aim to develop a theoretical analysis similar to QMIX for other CTDE algorithms. We would also like to carry out empirical evaluations for MAVEN when $z$ is continuous. To address the intractability introduced by the use of continuous latent variables, we propose the use of state-of-the-art methods from variational inference [24, 42, 41, 25]. Yet another interesting direction would be to condition the latent distribution on the joint state space at each time step and transmit it across the agents to get a low communication cost, centralised execution policy and compare its merits to existing methods [45, 11, 23].

# 8 Acknowledgements

AM is generously funded by the Oxford-Google DeepMind Graduate Scholarship and Drapers Scholarship. TR is funded by the Engineering and Physical Sciences Research Council (EP/M508111/1, EP/N509711/1). This project has received funding from the European Research Council under the European Union's Horizon 2020 research and innovation programme (grant agreement number 637713). The experiments were made possible by generous equipment grant by NVIDIA and cloud credit grant from Oracle Cloud Innovation Accelerator.

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
