[Supplementary Material]

# A  QMIX Architecture

Figure 8: The overall setup of QMIX, reproduced from the original paper [40] (a) Mixing network structure. In red are the hypernetworks that produce the weights and biases for mixing network layers shown in blue. (b) The overall QMIX architecture. (c) Agent network structure. Best viewed in colour.

# B  Proofs

## B.1  Uniform visitation

**Theorem 1.** *For $n$ player, $k \geq 3$ action matrix games ($|\mathcal{A}| = n, |U| = k$), under uniform visitation; $Q_{qmix}$ learns a $\delta$-suboptimal policy for any time horizon $T$, for any $0 < \delta \leq R\left[\sqrt{\frac{a(b+1)}{a+b}} - 1\right]$ for the payoff matrix given by the template below, where $b = \sum_{s=1}^{k-2}\binom{n+s-1}{s}$, $a = k^n - (b+1)$, $R > 0$:*

$$\begin{bmatrix} R+\delta & 0 & \dots & R \\ 0 & & & \ddots \\ \vdots & & \ddots & \vdots \\ R & \dots & & R \end{bmatrix}$$

*Proof.* For single state MDPs, under uniform visitation of the joint state-action space, QMIX can be seen as minimising the mean squared error between the actual $Q$-values and the monotonic projection $Q_{qmix}$. Using the symmetry of the problem and an exchange argument, it can be shown that only the monotonic projections of the following form need to be considered:

$$\begin{bmatrix} x_3 & x_2 & \dots & x_1 \\ x_2 & & & \ddots \\ \vdots & & \ddots & \vdots \\ x_1 & \dots & & x_1 \end{bmatrix}$$

where $X \triangleq (x_1, x_2, x_3)$. Consequently, there are two cases for the monotonic approximations. We refer to them as $M1$ and $M2$ corresponding to $x_1 \geq x_2 \geq x_3$ and $x_1 \leq x_2 \leq x_3$ cases respectively. The optimization problem for $M1$ is:

$$
\begin{aligned}
M1 : & \\
\underset{X}{\text{minimise}} \quad & a(x_1 - R)^2 + bx_2^2 + (x_3 - (R+\delta))^2 \\
s.t. \quad & x_2 - x_1 \leq 0 \\
& x_3 - x_2 \leq 0
\end{aligned}
$$

where $b = \sum_{s=1}^{k-2} \binom{n+s-1}{s}$, $a = k^n - (b+1)$ are the coefficients corresponding to the number of entries for the general $n$ player, $k$ action game (having $k^n$ entries). It is thus evident that the above problem is a quadratic program and is indeed convex [4] as the Hessian of the objective $diag(a, b, 1)$ is positive definite. The Largrangian is given by:

$$\mathcal{L}(X, \lambda_1, \lambda_2) = a(x_1 - R)^2 + bx_2^2 + (x_3 - (R + \delta))^2 + \lambda_1(x_2 - x_1) + \lambda_2(x_3 - x_2)$$

where $\lambda_1, \lambda_2$ are the dual variables. Moreover, the above problem also satisfies Slater's conditions which implies that KKT conditions are necessary and sufficient for finding the primal and dual optimal. By setting $\nabla_X \mathcal{L} = 0$, we get:

$$\lambda_1 = 2a(x_1 - R)$$
$$\lambda_2 = 2a(R + \delta - x_3)$$
$$x_2 = \frac{\lambda_2 - \lambda_1}{2b}$$

Using primal and dual feasibility constraints along with complementary slackness, we can see that $x_1 = R, x_2 = x_3 = \frac{R+\delta}{1+b}$ is an optimal solution to $M1$ for $\delta \leq bR$ with the optimal value for the problem as $OPT(M1) = \frac{b(R+\delta)^2}{b+1}$. By solving $M2$ in a similar way for the reversed primal constraints $x_2 - x_1 \geq 0, x_3 - x_2 \geq 0$, we see that an optimal assignment is $x_1 = x_2 = \frac{aR}{b+a}, x_3 = R + \delta$ with the optimal value given by $OPT(M2) = \frac{R^2 ab}{a+b}$. Note that the solution to $M1$ corresponds to the suboptimal policy of picking action corresponding to payoff $R$, whereas the solution to $M2$ corresponds to that of picking the optimal action with payoff $R + \delta$ (as QMIX picks the action corresponding to the maximal entry of a monotonic projection). For QMIX to learn the suboptimal policy corresponding to $M1$, we require that $OPT(M1) \leq OPT(M2)$. Consequently,

$$\frac{b(R + \delta)^2}{b + 1} \leq \frac{R^2 ab}{a + b}$$
$$\implies \delta \leq R\left[\sqrt{\frac{a(b+1)}{a+b}} - 1\right] \tag{2}$$

$\square$

### B.2  $\epsilon$-greedy visitation

**Theorem 2.** *For $n$ player, $k \geq 3$ action matrix games, under $\epsilon$-greedy visitation $\epsilon(t)$; $Q_{qmix}$ learns a $\delta$-suboptimal policy for any time horizon $T$ with probability $\geq 1 - \left(\exp(-\frac{T\upsilon^2}{2}) + (k^n - 1)\exp\left(-\frac{T\upsilon^2}{2(k^n-1)^2}\right)\right)$, for any $0 < \delta \leq R\left[\sqrt{a\left(\frac{\upsilon b}{2(1-\upsilon/2)(a+b)} + 1\right)} - 1\right]$ for the payoff matrix given by the template above, where $b = \sum_{s=1}^{k-2}\binom{n+s-1}{s}$, $a = k^n - (b+1)$, $R > 0$ and $\upsilon = \epsilon(T)$.*

*Proof.* Given the exploration schedule $\epsilon(t)$, let $\epsilon(T) = \upsilon$ (which is the minimum value since $\epsilon(t)$ is decreasing in $T$). We reuse the machinery introduced in Appendix B.1 and provide an analysis which is agnostic to the actions actually visited by considering the adversarial case for the maximum possible $\delta$ for which $Q_{qmix}$ fails. This happens precisely when QMIX is provided with the "best opportunity" for learning the optimal policy (so that it visits the optimal action with probability $1 - \epsilon(t), \forall t$). Therefore, the visitation frequencies we consider are : $\frac{T\upsilon}{k^n-1}$ for any suboptimal action and $T(1 - \upsilon)$ for the optimal action. To compute the upper bound on $\delta$, we modify the objective for the quadratic program in Appendix B.1 as $X^T diag(a', b', 1))X$ where $a' \leftarrow \frac{a\upsilon}{(1-\upsilon)(a+b)}, b' \leftarrow \frac{b\upsilon}{(1-\upsilon)(a+b)}$ in accordance with our visitations. Next, using the same reasoning as in Eq. (2), we get that QMIX learns the suboptimal policy for

$$0 < \delta \leq R\left[\sqrt{a\left(\frac{\upsilon b}{(1-\upsilon)(a+b)} + 1\right)} - 1\right]. \tag{3}$$

Note that the upper bound of $\delta$ in Eq. (3) is probabilistic in nature. Therefore, we provide a lower bound on the probability of this by considering the RHS of Eq. (3) with $\upsilon \leftarrow \upsilon/2$ and bounding the

probability of deviation from the worst case visitation frequencies. By making use of the Hoeffding's lemma, we derive that:

$$P\Big[\text{empirical frequency of optimal} - \upsilon T \geq \frac{T\upsilon}{2}\Big] \leq \exp(-\frac{T\upsilon^2}{2}),$$

$$P\Big[\text{empirical frequency of suboptimal} - \frac{T\upsilon}{k^n - 1} \leq -\frac{T\upsilon}{2(k^n - 1)}\Big] \leq \exp(-\frac{T\upsilon^2}{2(k^n - 1)^2}).$$

Finally, by using the union bound, we conclude that with probability $\geq 1 - \Big(\exp(-\frac{T\upsilon^2}{2}) + (k^n - 1)\exp(-\frac{T\upsilon^2}{2(k^n - 1)^2})\Big)$, QMIX fails to learn the optimal policy for

$$0 < \delta \leq R\left[\sqrt{a\Big(\frac{\upsilon b}{2(1 - \upsilon/2)(a + b)} + 1\Big)} - 1\right]$$

$\square$

### B.3 Variational Mutual Information lower bound

Let the posterior over $z$ be given by $\log(p(z|\sigma(\mathbf{u}, s)))$ and the variational approximation by $q_\upsilon(z|\sigma(\mathbf{u}, s)))$

$$
\begin{aligned}
\mathcal{J}_{MI} &= \mathcal{H}(\sigma(\mathbf{u}, s)) - \mathcal{H}(\sigma(\mathbf{u}, s)|z) \\
&= \mathcal{H}(z) - \mathcal{H}(z|\sigma(\mathbf{u}, s)) && \text{\{MI is symmetric\}} \\
&= \mathcal{H}(z) + \mathbb{E}_{\sigma(\mathbf{u}, s)}[\mathbb{E}_z[\log(p(z|\sigma(\mathbf{u}, s)))]] && \text{\{Def. conditional entropy\}} \\
&= \mathcal{H}(z) + \mathbb{E}_{\sigma(\mathbf{u}, s)}[\mathbb{E}_z[\log(p(z|\sigma(\mathbf{u}, s))) - \log(q_\upsilon(z|\sigma(\mathbf{u}, s)) + \log(q_\upsilon(z|\sigma(\mathbf{u}, s)))]] \\
&= \mathcal{H}(z) + \mathbb{E}_{\sigma(\mathbf{u}, s)}[\mathbb{E}_z[\log(q_\upsilon(z|\sigma(\mathbf{u}, s)))]] + \mathbb{E}_{\sigma(\mathbf{u}, s)}[KL(p(z|\sigma(\mathbf{u}, s))||q_\upsilon(z|\sigma(\mathbf{u}, s)))] \\
&\geq \mathcal{H}(z) + \mathbb{E}_{\sigma(\mathbf{u}, s), z}[\log(q_\upsilon(z|\sigma(\mathbf{u}, s)))] && \text{\{KL is non negative\}}
\end{aligned}
$$

## C Experimental Setup

### C.1 Architecture and Training

All agent are designed as Deep Recurrent Q-Networks [19]. At each time step, each agent network receives a local observation as input, which is fed to a $64$-dimensional fully-connected hidden layer, followed by a GRU recurrent layer and a fully-connected layer with $|U|$ outputs. To speed up the learning, all agent networks share the same set of parameters. A one-hot encoded agent id is concatenated to agent observations. The architectures for mixing and utility networks are the same as in [40].

For all experiments we update the target networks after every 200 episodes. We set $\gamma = 0.99$. The optimisation is conducted using RMSprop with a learning rate of $5 \times 10^{-4}$ and $\alpha = 0.99$ with no weight decay or momentum.

### C.1.1 SMAC

Exploration for QMIX is performed during training during which each agent executes $\epsilon$-greedy policy over its own actions. $\epsilon$ is annealed from $1.0$ to $0.05$ or $0.005$ over $50k$ time steps and is kept constant afterwards.

We utilise a replay buffer of the most recent 5000 environment steps. A single training step for a batch of size 32 entire episodes is performed after every episodes.

We set $Z = 16$ for all the experiments. We set $\lambda_{MI} = 0.001$ and $\lambda_{QL} = 1$. Unless otherwise mentioned, all MAVEN experiments use the trajectory-based MI loss. We use an entropy regularisation term with a coefficient of $0.001$ for the hierarchical policy. We set the final value of $\epsilon$ to $0.05$ for MAVEN ans QMIX.

(a) Varying the values for $Z$

(b) Policy returns for different $Z$

Figure 9: Performance with varying the number of latent variable categories

All SMAC experiments use the default reward and observation settings of the SMAC benchmark [43].

We run all methods for 10 million environmental steps. This takes approximately 36 hours on a NVIDIA GTX 1080Ti GPU for 12 random initializations.

### C.1.2 $m$-step matrix games

All methods anneal $\epsilon$ from 1 to 0.01 over 100 timesteps and keep it constant afterwards.

A single training step for a batch of size 32 is conducted after every episode.

All methods are run for $100k$ timesteps.

For MAVEN we set $Z = 16, \lambda_{MI} = 1, \lambda_{QL} = 1$ and use an entropy regularisation term with a coefficient of $0.001$ for the hierarchical policy.

### C.2 Additional plots & ablations

We also consider varying the number of categories for the discrete latent variable Fig. 9(a). While the number of categories loosely correlates with performance, it was not always the case. For `micro_corridor`, the results are inconclusive because they all use the same budget of gradient updates, yielding two opposing factors that cancel out (more $z$'s vs. less training per $z$). Fig. 9(b) gives the returns of the corresponding policies learnt.

(a) `2-corridors`

(b) `2s3z`

(c) `micro_corridor`

(d) `micro_focus`

Figure 10: Median test returns on SMAC scenarios.