[Reviews · NeurIPS 2019]

Reviewer 1



1. The proposed MAVEN method extends the idea of deep exploration via Bootstrapped DQN to the QMIX algorithm, aiming to provide committed exploration. One question is that this new exploration solves the sub-optimality issue of QMIX? Is the sub-optimality of value function decomposition methods (e.g., QMIX and VDN) intrinsic due to the full decomposition representation or due to the exploration? 2. Although it is shown that MAVEN significantly outperforms QMIX in the toy game problem, the experimental results in the SMAC domain seem not to demonstrate the expected significance of MAVEN over QMIX. The authors mentioned in the paper that the SMAC domain may not be ideal for testing exploration strategies, but it is highly desired for the authors to find an interesting challenging domain for evaluating the significance of MAVEN, as did by Bootstrapped DQN in the Atari games. In addition, it is also important to analyze the weakness of MAVEN. 3. Did the authors perform an ablation test for evaluating MAVEN against QMIX with each agent using Bootstrapped DQN? 4. How stable is the learning of MAVEN using the alternative training of the value function and the hierarchical control policy? 5. As shown in Figure 1, is the epsilon-greedy exploration still used in MAVEN? Some typos: Line 113: removing “a” Line 116: missing a “)” UPDATE: After reading the author's rebuttal, I have chosen to increase my score from 5 to 6 because the authors provide stronger results and partially address my Question 2.

Reviewer 2



The authors point out a shortcoming in a popular MARL algorithm and present a well-motivated solution. This much is fine. However, I think the uniqueness of both the problem and the solution were overemphasized. First, as far as I can tell, the inefficiency of exploration for QMIX boils down to the inefficiency of epsilon-greedy, a well-known shortcoming that plenty of people study in the single-agent q-learning domain. This could of course be exacerbated in the multi-agent setting (since the epsilon noise is added independently per agent), but the problem itself isn't new. So forgive me if I misunderstood something, but what is the point of section 3 beyond saying this? For me, this whole section could have been replaced with: "Epsilon-greedy is inefficient [citations]. The problem can be even worse in the multi-agent setting. [brief elaboration] Here we present a committed exploration approach for multi-agent q-learning." Second, the solution presented is essentially exactly DIAYN [Eysenbach et al 2018] (which itself was a generalization of variational intrinsic control [Gregor et al 2016]). Sure, DIAYN was applied in a different setting (unsupervised pre-training for RL), but the method of sampling a latent variable and encouraging behavioral trajectories to be as different as possible so as to allow the inference of that latent variable by a discriminator (as a variational approximation to optimizing mutual information) comes directly from that paper. They even had the same motivation of encouraging exploration. Given this, I find it unfair that exactly one sentence is devoted to DIAYN and VIC (lines 277-279). I think it would be more appropriate for this work to be highlighted prominently in the abstract, introduction, and section 4 on methodology (note that QMIX itself is highlighted in this way). ("MAVEN=QMIX+DIAYN" is the executive summary I wrote down for myself.) The Starcraft results also seem fine, but not so strong as it make it obvious that committed exploration is a crucial empirical improvement for QMIX - while MAVEN agents learn faster in 3s5z, the final performance looks the same; MAVEN agents seem to have less variability in final win rate on 5m_vs_6m; and QMIX actually seems to have better final performance on 10m_vs_11m. The results in figure 2 and 4 do however suggest that there may be scenarios where the advantage of MAVEN is higher. Minor comments: 1) line 64 and others: the subscript "qmix" should probably be wrapped in a "\text{}" 2) first eqn in section 3: inconsistency between using subscripts and superscripts, i.e. u_i and u^i 3) line 81: perhaps better phrased as: "the *best* action of agent i..." 4) line 86: u_n^i -> u_|U|^i? 5) line 87: I was confused by what "the set of all possible such orderings over the action-values" means. Besides a degeneracy when some of the Q values are identical, isn't there only one valid ordering? Or are you just trying to cover that degeneracy? 6) Definition 1: perhaps add an intuitive explanation, e.g. "Intuitively, a Q-function is non-monotonic if the ordering of best actions for agent i can be affected by the other agents action choices at that time step." 7) line 110: precise sequences -> precise sequence 8) line 131: for latent space policy -> for the latent space policy, missing space after period 9) line 162: should call the variational distribution a "discriminator" as it is introduced, both to help explain what role it is playing, and because this is done in Figure 1 without reference in the main text 10) line 174: sate -> state 11) Figure 2b: unexplained "[20]"s in legend can probably be removed 12) line 237: Fig 11 -> Fig 5 13) line 239+1: I think the ablation experiments were useful and interesting and should at least be summarized briefly in the main text. 14) Figure 5: should mention the number of training steps as one progresses left to right 15) line 289: we propose the use state -> we propose to use state OR we propose the use of state 16) line 290: condition latent distribution -> condition the latent distribution UPDATE: Thanks for your rebuttal. On my first point above, thanks for clarifying the strengths of your theoretical result; I underappreciated them on the first read-through. On my second point, thanks for clarifying the distinctions between VIC/DIAYN and your approach (though I do think you should include the discussion of the differences in your paper). Also, thanks for sharing the stronger empirical results. For all of these reasons, I've raised my score from a 5 to a 6.

Reviewer 3



Clarity: While the paper is readable, there are certainly rooms for improvements in the clarity of the paper. For example, the details of the algorithm in Section 4 is not straight forward and easy to follow, especially lines 146-182. Originality: The paper considers the MARL problem in which each agent has its own local observations, takes a local action and receive a joint reward, and the goal is to find the optimal action-value function. During the training each agent is allowed to access the action-observation of all agents, and the full state. It is shown that VDN and QMIX cannot represent the true optimal action-value function in all cases. In addition, for a fixed episode length $T$, it is proved that with increasing the exploration rate, decreases the probability of learning an optimal solution. Considering this, it is assumed that the lack of good exploration coupled with the representational limitations resulted to the sub-optimality of QMIX. To address this issues, an algorithm, multi-agent variational exploration (MAVEN), is proposed to resolve the limitation of monotonicity on QMIX's exploration. In this order a latent variable $z$ is introduced which is based on a stochastic variable $x \sim p(x)$ which $p$ is uniform or normal probability distribution. Function $g_\theta(x,s_0)$ returns $z$ and another neural network, called, hyper-net map $g_{\phi}(z,a)$, returns $W_{z,a}$. In parallel, to get the Q-function of each agent, a MLP gets $(o_i^t, a, u^{t-1}_i)$, pass the results in a GRU and the results of the GRU is mixed with $W_{z,a}$ to get $Q(\tau_i; z)$. The whole this block introduces parameters $\eta$. Then, the mixer network with parameters $\si$ obtains $Q_{tot}$. Also, a mutual information (MI) objective (what does "objective" mean here?) is also added into the model to encourage more exploration. Significance: There are several things to like about this paper: - The problem of safe RL is very important, of great interest to the community. - The way that the exploration is added in the model might be interesting for other to use in future. However, - I found the paper as a whole a little hard to follow specially in the algorithm side. - The experiments do not support the claim of the paper about the significance of the exploration.

[Author Response · NeurIPS 2019]

We thank all the reviewers for their feedback. All reviewers are concerned whether we substantially outperform QMIX.
Since StarCraft II experiments take a long time, we could not include all the results in the submission. However, we have
now obtained results for most of the SMAC maps [The StarCraft Multi-Agent Challenge, Samvelyan et al 2019], which
Samvelyan et al. have classified as Easy, Hard & Super Hard. We have also added a recent method QTRAN [Son et al,
ICML 2019] as another baseline. Results on several maps are shown below. Note in particular two of the **Super Hard**
maps in SMAC: Corridor (Fig. 1(a)) and 6h_vs_8z (Fig. 1(b)). The plots show that MAVEN performs substantially
better than all alternate approaches. We observe this trend on rest of **Super Hard** maps in SMAC with performance
similar to QMIX on **Hard** and **Easy** maps. **Thus MAVEN performs better as difficulty increases**. This, along with
the adaptation experiment in Fig 4(b) of the paper, strongly support the importance of the committed exploration offered
by MAVEN for decentralised MARL. We will include all the new results in the final version. Furthermore, QTRAN
does not yield satisfactory performance on most SMAC maps ($0\%$ win rate). The map on which it performs best is 2s3z
Fig. 1(c), an **Easy** map, where it is still worse than QMIX and MAVEN. We believe this is because QTRAN tries to
enforce optimal decentralisation using relaxed L2 penalties which are not sufficient for challenging domains.
In terms of theoretical contributions, **we are the first to analyse the effects of representational constraints on**
**exploration, a problem which is unique to decentralised MARL**. To this end, we also quantify the suboptimality of
QMIX, the current SOTA under *uniform* and $\epsilon$-*greedy* exploration. We next address the important individual reviewer

(a) corridor        (b) 6h_vs_8z        (c) 2s3z
comments. We refer to the specific comment by **C** followed by its number in their review:
**Reviewer 1 – C1:** Fully decomposed methods can in principle represent the optimal policy but poor exploration can
cause them to converge to suboptimal policies due to representational constraints like monotonicity. MAVEN solves
this by exploring different joint behaviour modes. **C3:** We provide that ablation for 3s5z map in App C.2 Fig 7(b) and
for the more illustrative corridor map here in Fig. 1(a) **red line MAVEN Uniform Z, MI=0**. **C4:** We did not encounter
any problem with stability as the parameters corresponding to the objectives are disjoint. **C5:** Yes, $\epsilon$-greedy is used to
regularise the training of neural nets involved in $Q$-value estimation.
**Reviewer 2 – C1:** We disagree with your evaluation about the uniqueness of the problem. **Inefficient exploration**
**hurts decentralised MARL agents, not just in the usual way of single agent RL, but more importantly it interacts**
**with the representational constraints necessary for decentralisation and can push the algorithm towards strictly**
**suboptimal policies**. While single agent RL can avoid convergence to suboptimal policies using various strategies like
increasing the exploration rate ($\epsilon$), ensuring optimality in the limit, this is not the case with decentralised MARL. As
**Theorem 2 shows, increased exploration can in fact reduce the chance of finding optimal behaviour**; this result
is in **stark contrast to single agent RL**. Existing methods for single-agent exploration do not address this problem.
**C2:** Not only is MAVEN different from DIAYN in the use case, it also enforces **action diversification** which means
the agents jointly learn to solve the task is many different ways; **this is how MAVEN prevents suboptimality from**
**representational constraints**, DIAYN is concerned only with discovering new states. Furthermore, DIAYN trains
on diversity rewards using RL whereas we train on them via gradient ascent. We will clarify these points in the final
version and add more discussion about DIAYN.
**Reviewer 3 –** Note that QTRAN was published 9 days before the submission deadline, so it was infeasible to
empirically test it. Now that we have tested QTRAN on SMAC, it clearly performs poorly compared to QMIX &
MAVEN on challenging domains like StarCraft II. In fact, there is **no empirical evidence that QTRAN can perform**
**well beyond toy domains**. The QTRAN paper evaluates it only on toy domains and uses 10 million training steps,
the same number we use for our much more complex domains. While Theorem 1 in the QTRAN paper guarantees
optimal decentralisation, it imposes $\mathcal{O}(|S||A|^n)$ **constraints on the optimisation problem** involved, where $|S|, |A|$
are the sizes of state and action spaces and $n$ is the number of agents. This is **computationally intractable** to solve in
discrete state-action spaces and is impossible given continuous state-action spaces. The authors of QTRAN propose
two algorithms (QTRAN-base & alt) which relax these constraints using two L2 penalties. Thus **QTRAN does not**
**overcome QMIX's limitations as it deviates from the exact solution**. We will discuss these points in the final version.
**C1:** As previously mentioned, there is no evidence that QTRAN is useful for challenging MARL domains but we think
the L2 penalties involved can still be included in MAVEN as auxiliary loss. **C2:** The hierarchical policy is on the latent
space variables which in turn control joint behaviour of agents for the entire episode. **C3:** The $m$-step matrix game is a
2-player multi-step game; the actions for the players are the row and column indices of the matrix to pick at each step;
the goal is to maximise total reward; we have marked the initial state in Fig 2(a). We will make sure to improve the
clarity of the algorithm section (4 Methodology).

[Meta-Review · NeurIPS 2019]

The paper presents a new exploration strategy for decentralized MARL that is based on a joint latent variable that is shared between the agent. This paper is a difficult case. While the theoretical insights concerning the difficulty of the exploration problem in decentralized MARL are insightful, the experimental results were not good enough in the original submission to convince the reviewers. The algorithm was only in one case considerably better than the competitor QMix and other baseline comparison were missing. However, in the rebuttal the authors provided much better results as well as additional comparison to Qtrans. The reviewers increased their score accordingly, the paper is now in an acceptable state.